# What Do Dental Students Know About E-Cigarettes? A Cross-Sectional Survey from One Palestinian Dental School

**DOI:** 10.3390/ijerph21121711

**Published:** 2024-12-23

**Authors:** Rola Muhammed Shadid, Reeta Alsaeed

**Affiliations:** 1Department of Prosthodontics, Faculty of Dentistry, Arab American University, Jenin 240, Palestine; reeta.alsaeed@aaup.edu; 2Private Practice, Jenin 240, Palestine

**Keywords:** e-cigarette, knowledge, belief, attitude, dental students, smoking, Palestine

## Abstract

Owing to the vital role played by dentists in patient education and due to the growing popularity of e-cigarette use among the younger population, this cross-sectional study aimed to assess the use, knowledge, beliefs, and attitudes toward e-cigarettes among dental students. A cross-sectional questionnaire was given between March and June of 2024 to undergraduate dental students in the Faculty of Dentistry at the Arab American University in Palestine. A 32-item questionnaire with five sections, namely demographics, smoking habits, knowledge, beliefs and attitude, and education about e-cigarettes was sent to all appropriate participants (N = 1050). The response rate was 31% (N = 325). About 11.7% of the respondents were current e-cigarette smokers. The two most commonly cited reasons for initial use of e-cigarettes were curiosity (52.8%) and a desire to quit conventional smoking (27.8%). The average knowledge score was 6.63 (3.08) out of 12, demonstrating a generally inadequate level of knowledge about e-cigarettes. Conventional cigarette smokers were more knowledgeable about e-cigarettes than non-smokers (OR = 1.928; 95% CI: 1.061–3.505; *p*-value = 0.031). The majority of students (92%) considered the level of education they received in the dental school about e-cigarettes as inadequate, and more than half (53.2%) received information from social media. In conclusion, the prevalence of e-cigarette use among Palestinian dental students is relatively high compared with that reported for other dental students worldwide. Since the students showed a generally inadequate level of knowledge about e-cigarettes, this study casts doubt on the competency of Palestinian dental students to offer cessation counseling, highlighting the necessity to revise the dental curricula to foster positive knowledge and belief conducive to ideal behaviors.

## 1. Introduction

The electronic cigarette (e-cigarette) was developed by a Chinese pharmacist, Hon Lik, in 2003 as a device for helping combustible cigarette smokers to gradually cease smoking [1]. E-cigarettes are electronic devices that heat liquid to produce an aerosol that usually consists of nicotine, a humectant like propylene glycol with or without glycerin, and a flavoring [1]. They encompass a mouthpiece, a cartridge for storing liquid, a heating element, a microprocessor, and a battery [1].

Nowadays, there is an increase in the usage and popularity of e-cigarettes worldwide, especially among the younger population [2]. It has been demonstrated that the prevalence of adults reporting their use has increased up to six-fold from 2010 to 2013 in the United States of America (USA) [3]. The Center for Disease Control and Prevention (CDC) reported that 8.1 million adults in the USA used them in 2018, while the highest proportion was among the younger ages of 18–24 years [4]. Other reports from Saudi Arabia demonstrated that the prevalence of their use ranged from 25 to 33.5% among the younger population [5,6]. Regarding Palestine, the Palestinian Central Bureau of Statistics reported that 4% of people aged 18 years and older in the West Bank used e-cigarettes in 2021 [7], while the prevalence of their use among Palestinian university students was 18.1%, as shown by a recent study [8].

Although the scientific community has revealed conflicting opinions about e-cigarettes, they could not be marketed as safe products. While the Royal College of Physicians stated that e-cigarettes’ health hazards are unlikely to surpass 5% of the hazards caused by combustible cigarettes [9], there is no evidence that e-cigarettes could be safer in the long term according to the European Respiratory Society (ERS) [10] and to the World Health Organization (WHO), who have raised warnings and concerns about their impact on health [11].

Several health hazards have been reported in the literature concerning the usage of e-cigarettes by adolescents and adults. Firstly, some studies showed that the liquids and aerosols in e-cigarettes might cause irritation and inflammation of airway mucosa that may lead to the development of chronic lung disease [12]. They might induce alterations of innate defense proteins in airway secretions, producing comparable and unique changes relative to combustible cigarette smoking [13]. They also may lead to immune suppression at the level of nasal mucosa [14]. Additionally, aerosols from e-cigarettes can comprise carcinogenic chemicals and small particles that can be inhaled deep into the lungs [15]. Nonetheless, in their review, Polosa et al. [16] revealed that the aforementioned studies [12,13,14] on the respiratory hazards of e-cigarettes suffered from some methodological flaws that should be taken into account, and they showed that normal usage of e-cigarettes led to less harm to the respiratory tract than combustible tobacco cigarettes [16]. However, dual use of electronic nicotine delivery systems and combustible tobacco cigarettes was significantly associated with a higher frequency of respiratory symptoms, compared with exclusive use of either one or noncurrent use at baseline [17].

Secondly, a review of preclinical and clinical studies showed that e-cigarette usage has been linked to oxidative stress, inflammation, and hemodynamic imbalance that might elevate the risk of cardiovascular disease [18]. However, this risk is less than that linked to combustible cigarette smoking [19]. Thirdly, most e-cigarettes contain nicotine, which has known negative health impacts. Nicotine is vastly addictive. It can affect the brain development of adolescents, and it may increase the risk for future addiction to other drugs [15]. Also, it is a health hazard for pregnant women due to its toxicity to developing fetuses [15].

Fourthly, a recent systematic review showed that e-cigarette usage has been associated with several oral health issues, including dental caries, periodontitis, gingival pain, and oral symptoms [20]. It also could lead to unintended injury, particularly among adult males aged ≥ 18 years. Though e-cigarette usage has led to fewer hospital-related injuries than combustible tobacco products, the sum of both sorts of injuries has stabilized rather than reduced over the prior decade [21]. Finally, some evidence has shown that youths who use e-cigarettes are more likely to use combustible tobacco cigarettes in the future, considering e-cigarette usage as a gateway to combustible cigarette smoking [22,23]. However, other researchers believe that the evidence of the “gateway effect” is limited, since it does not take into consideration the youths’ inherent propensity to use nicotine substances [24].

High-certainty evidence from a recent Cochrane review has indicated that nicotine e-cigarettes increase the quit rates of combustible cigarette smoking compared to nicotine replacement therapy [25]. Despite the documented short-term health hazards of e-cigarettes and the uncertainty about the long-term hazards, some authors and scientists advocate for the balancing of the convincing concerns about harm to youths with the probable benefits of assisting adults to quit combustible cigarette smoking [26]. For instance, health authorities in the United Kingdom and New Zealand officially recommend that adult smokers who are unable or unwilling to quit smoking switch to e-cigarettes to reduce health risks [27,28]. However, e-cigarettes should not be used by youths, young adults, or pregnant women due to potential health hazards [15].

Owing to all of the above-mentioned short-term hazards of e-cigarettes and the uncertainty of the long-term ones, 34 countries have prevented their sale, including Palestine, and 87 countries have permitted restricted sales [29].

Although there are several studies that have evaluated university students’ knowledge and belief about e-cigarettes [8,30,31,32,33,34,35,36,37,38,39,40], there are scarce data concerning dental students. A cross-sectional study from Saudi Arabia revealed a deficiency in dental students’ knowledge and confidence concerning e-cigarettes, highlighting the necessity for their education in dental school curricula [41]. A recent multinational survey revealed that the level of knowledge of dental students about e-cigarettes was unsatisfactory; however, they had positive beliefs and attitudes toward them [42].

Owing to the vital role played by dentists in patient education and in the execution of tobacco cessation programs, and because until now no study has been conducted in Palestine to assess dentists or dental students’ knowledge and beliefs about e-cigarettes, this cross-sectional study aimed to assess the use, knowledge, beliefs, and attitudes toward e-cigarettes among undergraduate dental students in the Faculty of Dentistry at the Arab American University (AAUP) in Palestine.

## 2. Methods

This cross-sectional study was directed and stated in accordance with CHERRIES guidelines [43] between March and June of 2024, targeting undergraduate dental students in the Faculty of Dentistry at AAUP. With a population size of 1050 dental students, an accepted margin of error of 5%, and a confidence interval of 95%, the minimum recommended sample size was determined as 282 participants according to an online sample size calculator (www.raosoft.com, accessed on: 18 December 2024). A convenient sample of 325 participants was enrolled in this cross-sectional study after they accepted to participate.

Inclusion criteria included being undergraduate dental students in the Faculty of Dentistry at the Arab American University, males or females, e-cigarette users or not, cigarette smokers or not, and having signed an informed consent. Students who did not accept to participate or did not complete the questionnaire, postgraduate students, teaching assistants, academic staff, or technicians, were excluded from the study.

Ethical approval was conceded from the institutional review board (IRB) at Arab American University (2024/A/2/N), and the study was conducted in accordance with the Declaration of Helsinki guidelines. The 32-item survey used in the present study was prepared after reviewing former studies [32,40,41]. The questionnaire was initially formed in English, assessed for content validity by an expert from the field, and then translated into Arabic. It was evaluated for clarity and simplicity by conducting a pilot study on 20 students who did not participate in the original questionnaire.

The internal consistency between the survey’s items of the 325 participants was evaluated using the coefficient alpha “Cronbach’s alpha”. A Cronbach α = 0.848 for knowledge items and α = 0.729 for belief and attitude items were obtained, revealing satisfactory internal consistency [44].

Participants were recruited using the convenience sampling technique (snowball technique). Google Drive was used to make the online questionnaire, and a link was sent to all appropriate participants via email and via closed groups on social media. The questionnaire was sent with a cover letter demonstrating the study objectives, the methods, and confirming that participation in this questionnaire was voluntary and anonymous, and all collected data would be kept securely for questionnaire purposes only. All respondents were requested to provide informed consent, and the survey was estimated to take 6–7 min.

The questionnaire comprised 32 close-ended questions that were divided into 5 categories: demographics, smoking habits, knowledge, belief and attitude, and education. The first category included data about age, sex, and year of study. The smoking habits category involved 7 items. The knowledge category consisted of 12 items with 3 response options (true, false, do not know). The “do not know” response was regarded as an “incorrect” answer. A score of “1” was assumed for each correct answer; therefore, the best knowledge score was 12. While 7.6 (63.6%) represented the cut-off score, >7.6 was considered knowledgeable and ≤7.6 as non-knowledgeable [40]. The fourth category of belief and attitude consisted of 7 items. Participants responded to belief statements using “strongly agree”, “agree”, “neutral”, “disagree”, and “strongly disagree”. Finally, the fifth category of education included 3 items.

### Statistical Analysis

Responses were gathered using a Google Drive Excel document, and data were analyzed statistically using the Statistical Package for Social Sciences (SPSS), version 22.0. Means and standard deviations were measured for all continuous variables and percentages for all qualitative variables. Univariate analysis was executed using Chi-square and Fisher Exact tests, and variables with a *p*-value less than 0.05 were entered in multivariate analysis. Independent factors influencing students’ knowledge were determined using binary logistic regression, and both odds ratio (OR) and 95% confidence interval (95% CI) were calculated. The level of statistical significance was set at *p* < 0.05.

## 3. Results

Of the 1050 dental students who were invited to participate, 325 filled out the questionnaire, with a response rate of 31%; 61.8% of the respondents were female and 91.7% were ≤23 years old (Table 1).

### 3.1. Smoking Habits Among Participants

A total of 18.4% of the students were current conventional tobacco smokers, while only 11.7% were current e-cigarette users. About half of e-cigarette users (54.4%) had been using e-cigarettes for 1–3 years. Among e-cigarette users, 51.1% reported daily use, and 68.1% spent 30 min or more in each vaping session. A high level of awareness of e-cigarettes was observed among dental students, with an impressive 98.5% of participants acknowledging familiarity with it. Friends (58.5%) were the most common awareness source, followed by social media (20%) and family members (14.2%). Only 1.5% of participants had never heard of e-cigarettes before. Participants cited various reasons for their initial trial of e-cigarettes, including curiosity (52.8%), a desire to quit conventional smoking (27.8%), the belief that e-cigarettes are safer than conventional cigarettes (8.3%), a recommendation from a friend (6.9%), and others (Table 2).

Regarding the factors that were related to e-cigarette usage, age, gender, and year of study did not have a significant relationship with its usage; however, conventional smoking status was significantly related (*p* < 0.05). Around 34.8% of conventional cigarette smokers (current/ex-smokers) have used e-cigarettes, while just 11.7% of non-conventional smokers have used e-cigarettes (Table 3).

### 3.2. Students’ Knowledge About E-Cigarettes

The average knowledge total score was 6.63 (3.08) out of 12. Less than half (45.5%) of the respondents recognized that e-cigarettes are reusable. Around 51.4% misconceived that all e-cigarettes contain natural ingredients, while only 32.9% recognized that they are regulated as tobacco products. Moreover, 70.8% were knowledgeable that they contain carcinogenic elements, and 46.8% understood that they can cause second-hand nicotine exposure. Most participants recognized that their aerosols could increase heart rate or arterial stiffness (77.2%), raise blood pressure (73.2%), cause airway obstruction (83.1%), and adversely affect oral health (80%) (Table 4).

Age, sex, year of study, and usage status of conventional cigarettes and e-cigarettes were factors that were examined as predictors for students’ knowledge about e-cigarettes. The multivariate analysis showed that conventional cigarette smoking was significantly associated with knowledge about e-cigarettes (Table 5). Current conventional cigarette smokers were more knowledgeable about e-cigarettes than non-smokers (OR = 1.928; 95% CI: 1.061–3.505; *p*-value = 0.031).

### 3.3. Students’ Beliesf and Attitudes Toward E-Cigarettes

Around 18.5% of participants believed that e-cigarettes are less harmful than conventional cigarettes. Over a quarter (26.4%) agreed that they are less addictive. Only 15.4% considered their use as an effective method for quitting conventional smoking. When asked if they would recommend e-cigarettes to friends or family members, 18.5% agreed, 72.9% disagreed, and 8.6% were “neutral”. Regarding the feeling of confidence to discuss conventional cigarettes or e-cigarettes with their patients, (57.5, 56.9%) felt confident, (30.2, 29.2%) reported lack of confidence, and (12.3, 13.9%) were neutral, respectively (Table 6).

Compared to those who used e-cigarettes, participants who did not use e-cigarettes were significantly more likely to contradict the belief that e-cigarettes are less harmful, less addictive, or an effective way for smoking cessation; therefore, they would not recommend e-cigarettes to friends or family members as a smoking cessation tool (*p* < 0.05) (Table 7).

### 3.4. Education About E-Cigarettes

Regarding the education the students received about e-cigarettes in their dental school, the majority of the students (92%) considered the level of education as inadequate, and more than half of them (53.2%) received information about e-cigarettes from social media. Other reported sources of information were online advertising, billboards and/or public signs, newspapers or magazines, television advertising, radio advertising, and others (Table 8).

## 4. Discussion

This cross-sectional study sought to assess the use, knowledge, beliefs, and attitudes toward e-cigarettes among undergraduate dental students in Palestine and to investigate if the students’ demographics or smoking status would have an effect.

A high level of awareness of e-cigarettes was observed among dental students, with an impressive 98.5% of participants acknowledging familiarity with e-cigarettes. This is in accordance with findings from other studies surveying Jordanian university [40] and Italian nursing students [36].

The current study indicated that the majority of students were non-smokers; just 11.7% were using e-cigarettes and 18.4% were engaging in conventional cigarette smoking. This finding is comparable to what was reported in a Jordanian study that showed 10.5% prevalence of e-cigarette usage among university, though non-dental, students [40]. However, the prevalence in our study is lower than what was reported from a recent study conducted on a larger sample of Palestinian university students from five different universities, in which the prevalence of e-cigarette usage was 18.1% [8]. Although that study had a comparable female-to-male ratio to our study, it surveyed a larger sample from different universities with different study fields rather than exclusively dentistry. This could explain the differences in the prevalence of e-cigarette usage.

On the contrary, our finding is higher than that of a Saudi survey of dental students that showed 5.7% prevalence in 2020 [41], and also higher than that of a multinational survey of dental students that revealed just 4.5% prevalence of e-cigarette usage in 2022 [42]. Eleven countries were included in this multinational survey (Croatia, Iraq, Jordan, Kuwait, Lebanon, Malaysia, Nigeria, Saudi Arabia, South Africa, Turkey, and Yemen) [42]. Though a much higher male-to-female ratio was in the Saudi study in comparison to that of our study [41], and a comparable female-to-male ratio was in the multinational study [42] in relation to ours, the higher prevalence of e-cigarette use in our study could be explained by the worldwide growth in e-cigarette use, the differences in educational, social, and economic levels, and/or regional disparity between students in different studies.

Regarding the reported reasons to use e-cigarettes for the first time in the current study, curiosity (52.8%) was the most common reason, potentially driven by the appealing marketing strategies of e-cigarette producers. The second most cited reason, at 27.8%, was a desire to quit conventional cigarette smoking. Comparable results for using e-cigarettes as an aid in smoking cessation were reported from studies surveying Jordanian university students [40], Polish medical students [30], American health professional students [31], and Nepali university students [45]. Nonetheless, a significant convergence in the usage of conventional cigarettes and e-cigarettes was reported in the current study and in previous studies [33,40], suggesting that using e-cigarettes may exacerbate, rather than alleviate, the issue of tobacco use among the young adults.

The current study demonstrated a generally inadequate level of knowledge about e-cigarettes, with an average score of 6.63 out of 12. Though different items, demographics, sample size, and scales were used in other studies, this score is lower than what was reported in a Palestinian study of university students, which was 4.47 out of 7 [8]; however, it is higher than what was stated in a multinational survey [42] of dental students with an average score of 2.9 out of 7, and also higher than that reported from a study surveying Jordanian university students [40], which demonstrated that the average knowledge score across both medical and non-medical students fell under 50% correct. Additional outcomes across studies from the USA [31] and Saudi Arabia [34,35] echo our results, underscoring the necessity to notify college students—regardless of their field of study—that all forms of tobacco consumption are deleterious, with e-cigarettes potentially posing a greater addiction risk than conventional cigarettes [46].

Such a lack of awareness, particularly among future dental professionals, has potentially negative implications for their role modeling with patients. A recent survey of dental students demonstrated that Malaysian and Turkish dental students’ knowledge levels were statistically superior compared to their international counterparts. The authors attributed this result to Malaysia’s integration of smoking cessation programs into some dental school curricula starting from the second year through the fifth, with reinforced clinical training [42]. Recognizing the essential need for comprehensive understanding regarding e-cigarettes, Briggs et al. [47] provided contemporary information about their detrimental effects, counseling recommendations, and potential regulations after conducting surveys among dental professionals, dental associations, and regulatory bodies.

Regarding the belief of harmfulness and addictiveness of e-cigarettes compared with conventional cigarettes, 18.5% and 26.4% of the surveyed dental students believed that e-cigarettes are less harmful and less addictive, respectively. Recent findings from a multinational study revealed that 33.1% of dental students believed that e-cigarettes are less harmful to health, and 51.9% recognized the addictiveness of e-cigarettes [42]. In addition, 51.7% of Saudi students believed that e-cigarettes are less addictive than conventional cigarette smoking [38], whereas 96.5% [39] and 85.6% [32] of American medical students acknowledged the health dangers and addictive nature of e-cigarette usage, respectively. Notably, findings from a recent study indicated that a significant number of American adults view e-cigarettes as equally or more detrimental than conventional cigarettes, with this sentiment growing markedly between 2012 to 2015 [48] and from 2012 and 2017 [49].

Concerning students’ beliefs also, only 15.4% of our students considered e-cigarette usage as an effective method for quitting conventional smoking, and accordingly, just 18.5% would recommend e-cigarettes to friends or family members as an aid for conventional smoking cessation. In a divergence from our results, 31.6% of multinational dental students [42] and 59% of Saudi medical students [34] believed so. Likewise, French laypeople shared affirmative beliefs about e-cigarettes as an aid for smoking cessation [50]. However, an Egyptian study showed comparable results to ours, with only 13% of healthcare providers and 20% of the general public believing in the effectiveness of e-cigarettes for smoking cessation [51].

Regarding the feeling of confidence to discuss the detrimental effects of e-cigarettes with their patients, 56.9% felt confident. Comparable to our findings, 48.1% of dental students in a multinational survey reported so [42]. Conversely, a majority of dental students in one Saudi dental school did not feel confident about their ability to discuss e-cigarettes with their patients [41].

In assessing whether students’ demographics or cigarette usage status affected the knowledge about e-cigarettes, the present study showed that the knowledge about e-cigarettes was significantly associated with current use of conventional cigarettes, as current conventional cigarette smokers were more knowledgeable about e-cigarettes than non-smokers. This result is in agreement with that of another study [40].

Concerning the education gained from their dental school about e-cigarettes, the majority of the surveyed dental students (92%) considered the level of education as inadequate; this finding is in agreement with that from a survey of Saudi dental students [41]. However, more than half of the respondents (53.2%) in the current study received information about e-cigarettes from social media, followed by online advertising, billboards and/or public signs, and others; a finding that aligns with observations made in prior research [40,52]. While social media platforms offer a means, they should not be seen as a substitute for formal education due to the presence of unsubstantiated or poorly evidenced information on these networks [41]. Although controlling or regulating the quality of marketing information disseminated on these social networks can be challenging [53], social media can complement professional guidance in educating various audiences, including patients, students, dentists, and physicians regarding e-cigarettes [41]. However, the most important is to integrate comprehensive and adequate information into the curricula of dental and medical schools [41,42]. Further, creating national intervention programs aiming to raise awareness among university students regarding the potential dangers of e-cigarettes is essential [50].

The current study gains its significance from being a cross-sectional, affordable, and instantaneous tool that gives insight into the competency of undergraduate dental curricula regarding e-cigarette education. This study is pioneering in documenting the varied dimensions of e-cigarette usage in Palestinian dental schools where such data are scarce. The study is the first to offer an assessment of e-cigarette usage, knowledge, beliefs, and attitudes among dental students in Palestine. Since it is likely that numerous dental practitioners will come across individuals who use e-cigarettes [54], it is essential to receive adequate education about e-cigarettes as part of their training programs and through awareness initiatives.

Nevertheless, caution should be taken when considering the results of the present study due to some certain limitations. Initially, this study included only dental students in the Faculty of Dentistry at AAUP; this may restrict the generalizability of the results to all dental students in Palestine. Another limitation is the study’s cross-sectional design and reliance on self-reported questionnaires, methods often criticized for their susceptibility to recall bias and subjective responses. Additionally, collecting data online may lead to the inadvertent exclusion of students lacking internet access, potentially leaving out individuals from lower socioeconomic groups, which could skew the results. Furthermore, the relatively low response rate (31%) could announce a nonresponse bias that should be borne in mind. Lastly, failing to investigate what motivates dental students to use conventional and/or e-cigarettes omits valuable insights that could inform preventive or cessation programs.

It is highly recommended to conduct additional large-scale, meticulously structured studies that encompass a broader range of dental students and dentists. The inclusion of objective outcomes, such as biological parameters, should be emphasized. Further research on other groups of health students, including medical and nursing students, is also advised. When considering curriculum reforms, conducting pre- and post-intervention studies can provide insights into the beneficial impacts and the needed adjustments.

## 5. Conclusions

The prevalence of e-cigarette use among Palestinian dental students is relatively high compared with that reported for other dental students worldwide. Despite the majority of the participants acknowledging familiarity with e-cigarettes, the level of knowledge was generally inadequate. Students who smoked conventional cigarettes were more knowledgeable about e-cigarettes than non-smokers. More than half of the respondents in the current study received information about e-cigarettes from social media, and nearly one third of those students cited “a desire to quit conventional smoking” as a reason for e-cigarette usage.

Since the prevalence of cigarette smoking, the most concerning public health threat, is also high and even higher (18.4%) than e-cigarette usage (11.7%), besides the doubt cast by this study on the competency of Palestinian dental students to offer cessation counseling, it is imperative that the dental curricula in Palestinian dental schools be revised to foster positive knowledge and beliefs toward the use of either form of cigarettes. Such reforms promise to nurture conviction among dental students, to dispel the prevalent myths concerning cigarette use, and to prepare them to be exemplars within their communities. Additionally, this will equip them with the necessary skills for more informed clinical judgments and effective patient advice concerning both conventional and e-cigarette usage.

## Figures and Tables

**Table 1 ijerph-21-01711-t001:** Demographics and characteristics of participants (N = 325).

Variable	N (%)
**Age**	
18–20 years	97 (29.8)
21–23 years	201 (61.9)
≥24 years	27 (8.3)
**Sex**	
Female	201 (61.8)
Male	124 (38.2)
**Year of study**	
1st & 2nd	103 (31.7)
3rd & 4th	133 (40.9)
5th	89 (27.4)

**Table 2 ijerph-21-01711-t002:** Smoking habits of participants (N = 325).

Smoking Habit	N (%)
**Do you smoke conventional cigarettes?**	
No (never smoked)	256 (78.8)
Yes (current smoker)	60 (18.4)
ex-smoker	9 (2.8)
**Do you use e-cigarettes?**	
No (never used)	271 (83.4)
Yes (current user)	38 (11.7)
ex-user	16 (4.9)
**Where have you heard about e-cigarettes?**	
Website advertisements	19 (5.8)
Social media	65 (20.0)
Family	46 (14.2)
Friends	190 (58.5)
I have not heard of it before	5 (1.5)
**Since when have you used e-cigarettes?**	
<1 year	15 (32.6)
1–3 years	25 (54.4)
>3 years	6 (13)
**How many times do you use e-cigarettes?**	
Daily	23 (51.1)
Twice weekly	6 (13.3)
Once weekly	1 (2.2)
Twice a month	5 (11.1)
Once a month	10 (22.2)
**For how many minutes do you use e-cigarettes?**	
<30 min	15 (31.9)
30–60 min	15 (31.9)
>60 min	17 (36.2)
**Reasons for using e-cigarettes for the first time.**	
To stop conventional cigarette smoking	20 (27.8)
To reduce nicotine exposure	6 (8.3)
Curiosity	38 (52.8)
Recommendation from friend	5 (6.9)
It is permissible in areas where smoking is prohibited	3 (4.2)

**Table 3 ijerph-21-01711-t003:** Factors associated with e-cigarette use among dental students of AAUP (N = 325).

Variable	Never Used E-Cigarettes (n = 271)	Current E-Cigarette User/Ex-User (n = 54)	*p*-Value
**Age**			0.752 ‡
18–20 years	83 (85.6)	14 (14.4)	
21–23 years	166 (82.6)	35 (17.4)	
≥24 years	22 (81.5)	5 (18.5)	
**Sex**			0.098 †
Female	173 (86.1)	28 (13.9)	
Male	98 (79.0)	26 (21.0)	
**Year of study**			0.370 †
1st & 2nd	86 (83.5)	17 (16.5)	
3rd & 4th	107 (80.5)	26 (19.5)	
5th	78 (87.6)	11 (12.4)	
**Conventional smoking status**			<0.001 †*
Current/ex-smoker	45 (65.2)	24 (34.8)	
Never smoked	226 (88.3)	30 (11.7)	

† Chi-square test ‡ Fisher Exact test. * *p*-value *<* 0.05 was considered significant.

**Table 4 ijerph-21-01711-t004:** Knowledge of participants about e-cigarettes and knowledge score (N = 325).

Question		Survey Results N (%)	
E-cigarettes are reusable		148 (45.5) “Yes” answer	
All e-cigarettes contain natural substances		158 (48.6) “No” answer	
All e-cigarettes do not contain nicotine		198 (60.9) “No” answer	
E-cigarettes operate on batteries that do not burn tobacco		68 (20.9) “Yes” answer	
Overall cost of e-cigarettes is less than that of conventional cigarettes		74 (22.8) “No” answer	
E-cigarettes are a source of second-hand exposure to nicotine		152 (46.8) “Yes” answer	
E-cigarettes are regulated as tobacco products		107 (32.9) “Yes” answer	
E-cigarettes do not contain carcinogenic ingredients		230 (70.8) “No” answer	
E-cigarette aerosols increases the heart rate/arterial stiffness		251 (77.2) “Yes” answer	
E-cigarette aerosols increases the blood pressure		238 (73.2) “Yes” answer	
E-cigarette aerosols induces obstruction of conducting airways		270 (83.1) “Yes” answer	
E-cigarette aerosols negatively affects oral heath		260 (80.0) “Yes” answer	
**Knowledge score**	Not knowledgeable (0–7.6)	Knowledgeable (>7.6)	Mean (SD)
**N (%)**	160 (49.2)	165 (50.8)	6.63 (3.08)

*SD* standard deviation.

**Table 5 ijerph-21-01711-t005:** Predictors of knowledge about e-cigarette use (N = 325).

Variable	OR (95% CI)	*p*-Value
**Conventional smoking status**		
Never smoked	Reference	
Current	1.928 (1.061–3.505)	0.031 *
Ex-smoker	6.343 (0.752–53.513)	0.090
**E-cigarette usage status**		
Never used	Reference	
Current	2.021 (0.958–4.264)	0.065
Ex-user	2.479 (0.746–8.238)	0.138

* *p*-value *<* 0.05 was considered significant.

**Table 6 ijerph-21-01711-t006:** Beliefs and attitudes toward e-cigarettes (N = 325).

Question	Agree N (%)	Disagree N (%)	Neutral N (%)
Using e-cigarettes is less dangerous than smoking conventional cigarettes	60 (18.5)	198 (60.9)	67 (20.6)
Using e-cigarettes is less addictive than smoking conventional cigarette	86 (26.4)	178 (54.8)	61 (18.8)
E-cigarette aerosols are less dangerous than cigarette aerosols	77 (23.7)	166 (51.1)	82 (25.2)
Using e-cigarettes is an effective way for smoking cessation	50 (15.4)	198 (60.9)	77 (23.7)
I will recommend use of e-cigarettes to my friends and family members as a smoking cessation method	60 (18.5)	237 (72.9)	28 (8.6)
As a student, I feel confident about my ability to discuss traditional cigarette use with my patients	187 (57.5)	98 (30.2)	40 (12.3)
As a student, I feel confident about my ability to discuss e-cigarette use with my patients.	185 (56.9)	95 (29.2)	45 (13.9)

**Table 7 ijerph-21-01711-t007:** Beliefs and attitudes of participants toward e-cigarettes by e-cigarette use status (N = 325).

Belief/Attitude Item	E-Cigarette Usage Status N (%)		*p* Value †
	No	Yes/Ex-User	
**Using e-cigarettes is less dangerous than smoking conventional cigarettes**			<0.001 *
Disagree	180 (66.4)	18 (33.3)	
Agree/neutral	91 (33.6)	36 (66.7)	
**Using e-cigarettes is less addictive than smoking conventional cigarette**			0.002 *
Disagree	159 (58.7)	19 (35.2)	
Agree/neutral	112 (41.3)	35 (64.8)	
**E-cigarette aerosols are less dangerous than cigarette aerosols**			0.001 *
Disagree	150 (55.4)	16 (29.6)	
Agree/neutral	121 (44.6)	38 (70.4)	
**Using e-cigarette is an effective way for smoking cessation**			<0.001 *
Disagree	180 (66.4)	18 (33.3)	
Agree/neutral	91 (33.6)	36 (66.7)	
**I will recommend use of e-cigarettes to my friends and family members as a smoking cessation method**			<0.001 *
Disagree	212 (78.2)	25 (46.3)	
Agree/neutral	59 (21.8)	29 (53.7)	
**As a student, I feel confident about my ability to discuss conventional cigarette use with my patient**			0.747
Agree	157 (57.9)	30 (55.6)	
Disagree/neutral	114 (42.1)	24 (44.4)	
**As a student, I feel confident about my ability to discuss e-cigarette use with my patient**			0.410
Agree	157 (57.9)	28 (51.9)	
Disagree/neutral	114 (42.1)	26 (48.1)	

Note: The percentage is calculated within the rows. † Chi-square test. * *p*-value *<* 0.05 was considered significant.

**Table 8 ijerph-21-01711-t008:** Education of participants about e-cigarettes (N = 325).

Question	N (%)
1. Do you receive any education about e-cigarettes in your college?	45 (13.8) “Yes” answer
2. I feel I have received sufficient education about e-cigarettes in the dental curriculum	26 (8.0) “Yes” answer
3. Where have you received information about e-cigarettes outside of dental school?	
Social media	173 (53.2)
Online advertising	44 (13.5)
Television advertising	3 (0.9)
Radio advertising	1 (0.3)
Billboards and/or public signs	6 (1.9)
Newspapers or magazines	6 (1.9)
Others	92 (28.3)

## Data Availability

The data that support the findings of this study are submitted with the manuscript.

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
