# Peer review of "What Do Dental Students Know About E-Cigarettes? A Cross-Sectional Survey from One Palestinian Dental School"

_ijerph, 2024, doi:10.3390/ijerph21121711_

Round 1

Reviewer 1 Report (Previous Reviewer 3)

Comments and Suggestions for Authors

The authors have significantly improved the quality of the manuscript. However, some gaps are still being apparent, which could be summarized as follows:

1. The most important concern is related to the cut-off of knowledge, which is determined according to the study of Al-Sawalha et al., 2021. In fact, the authors of this study used a cutoff of 7 out of 11, which represents 63.63% (even though they did not explain why they used this scale). With this %, your cut-off will be 7.63, not 7.2 (63.63, not 60%). Revise accordingly (even the results will not change).

2. The second concern is related to the predictors of knowledge. In fact, the authors should first provide the results of the univariate analysis and then decide about the p value that should allow to include a factor in the multivariate analysis ( Al-Sawalha  et al, 2021 used a p value less than 0.25 for example, what is the case of your study?). These factors should also be discussed.

3. The other concerns are related to the multiple grammatical and spelling errors that should be revised:

Lines 121-122: "Provided…1050". Revise the sentence

Table 3: Correct the expression " E-cigarette never user".

Revise also the numbers and the spaces between brackets.

Line 203-204: delete the sentence " demonstrating….about e-cigarette".

Table 4: improve the quality of the added part (Non-knowledgeable (0-7.2) in one line..).

Revise all the titles of the figures

Line 256: revise the expression " Non-users of e-cigarette compared to e-cigarette users."

Line 248: add "a" before Jordan

Line 288-291: reformulate

Line 301: How do you explain this high rate in Palestine?

Dear Reviewer 1,

Firstly, I would like to thank you for your valuable comments that significantly improved the manuscript. Please find the responses for each point.

Firstly, I would like to thank you for your valuable comments that significantly improved the manuscript. Please find the responses for each point.

1.The authors have significantly improved the quality of the manuscript. However, some gaps are still being apparent, which could be summarized as follows:

The most important concern is related to the cut-off of knowledge, which is determined according to the study of Al-Sawalha et al., 2021. In fact, the authors of this study used a cutoff of 7 out of 11, which represents 63.63% (even though they did not explain why they used this scale). With this %, your cut-off will be 7.63, not 7.2 (63.63, not 60%). Revise accordingly (even the results will not change).

Response: Thank you. Done as highlighted. L156-157

  1. The second concern is related to the predictors of knowledge. In fact, the authors should first provide the results of the univariate analysis and then decide about the p value that should allow to include a factor in the multivariate analysis (Al-Sawalha  et al, 2021 used a p value less than 0.25 for example, what is the case of your study?). These factors should also be discussed.

Response: Thank you. Done as highlighted. L165-166. L228-231. Table 5 modified.

  1. The other concerns are related to the multiple grammatical and spelling errors that should be revised:

Response: Thank you. Done as highlighted.

Lines 121-122: "Provided…1050". Revise the sentence

Response: Thank you. Done as highlighted. L121-123

Table 3: Correct the expression " E-cigarette never user".

Response: Thank you. Done as highlighted.

Revise also the numbers and the spaces between brackets.

Response: Thank you. Done.

Line 203-204: delete the sentence " demonstrating….about e-cigarette".

Response: Thank you. Done.

Table 4: improve the quality of the added part (Non-knowledgeable (0-7.2) in one line..).

Response: Thank you. Done as highlighted.

Revise all the titles of the figures

Response: Thank you. Done as highlighted.

Line 256: revise the expression " Non-users of e-cigarette compared to e-cigarette users."

Response: Thank you. Done as highlighted. L250

Line 248: add "a" before Jordan

Response: Thank you. Done as highlighted.

Line 288-291: reformulate

Response: Thank you. Done as highlighted. L283-285

Line 301: How do you explain this high rate in Palestine?

Response: The higher rate in our study could be explained by that e-cigarette use is growing and gaining more popularity worldwide, as the compared studies [41,42] were performed in 2020 and 2022, respectively. L294-296

Reviewer 2 Report (Previous Reviewer 2)

Comments and Suggestions for Authors

This is the second round of reviewing. The authors have addressed al the points of criticism I raised. While I do not agree with all their arguments, I believe the article can provide useful information to the readership, therefore I recommend its publication as it stands.

Author Response

Dear Reviewer 2,

I would like to thank you for your valuable comments that really improved the manuscript.

Sincerely,

Corresponding author.

This manuscript is a resubmission of an earlier submission. The following is a list of the peer review reports and author responses from that submission.

Round 1

Reviewer 1 Report

Comments and Suggestions for Authors

Reviewer 2 Report

Comments and Suggestions for Authors

The report is contained in an attached pdf file

Reviewer 3 Report

Comments and Suggestions for Authors

I would thank the authors for this interesting work. After reviewing I found some gaps that bothered me:

1. The first concern is related to the inclusion criteria and the sample characteristics. In fact, including females from an Arabic society to assess the smoking rate seems to be misleading since it is well known that females in these countries are generally non smokers and are generally not allowed to smoke away. The results are really questionable and could also be related to social desirability (biases). The fact that most of the participants were female could affect the results. In addition you should take this point into consideration when comparing with other studies in other countries.  

You should also be aware when comparing the level of knowledge (in the discussion), we compare the level of knowledge just if we have the same scale and/or the same items.

I have also other "minor" comments:

Reformulate the introduction of the abstract and add a conclusion.

Complete the sentence of lines 25-26 "despite…E-cigarettes"

You should reorder your ideas in the introduction: you should begin with the definition of –cigarettes (lines 46-53) and then you provide the epidemiology and the danger….

You should avoid to repeat the ideas (ex: sentence of line 84 is provided in the first line of the introduction).

You should provide the context of E-cigarette in Palestine and if no in other neighboring Arabic countries

Methods:

You should provide statistical analysis in this part not in the results.

Results:

Some data are missed since the number that should be provided for Smoker/Ex-smoker should be 54.

Try to add subtitles and reorganize your results by putting the tables just after the corresponding results.

Try to avoid the use of introductive sentences (exp: lines 90-92, 215-217, …)

Discussion

You should summarize the discussion by discussing just the most important results. In addition the number of used studies is very limited(3 or 4 only). However, You should be aware when comparing due to difference in demographic characteristics or in the used items and scales.

In line 263-265: how did you explain this high rate in Palestine? (even you included females)

Complete the first sentence of the conclusion.

At last, the manuscript require an English language editing.

Comments on the Quality of English Language

Moderate Editing required

Round 2

Reviewer 2 Report

Comments and Suggestions for Authors

What do dental students know about E-cigarette? A cross-sectional survey from one Palestinian dental school,
Rola Muhammed Shadid and Reeta Alsaeed,
ijerph-3221596
Second revision

The authors have considerably improved the manuscript. However, before acceptance they need to address the following minor points:

(1) In my previous report I requested the authors to cite reference [16] as a critique of studies on respiratory effects of e-cigarette usage (the authors' references [12,13,14,17]).

Polosa R, O'Leary R, Tashkin D, Emma R, Caruso M. The effect of e-cigarette aerosol emissions on respiratory health: a narrative review. Expert Rev Respir Med. 2019 Sep;13(9):899-915. doi: 10.1080/17476348.2019.1649146.

The authors added the following text commenting on this reference

"While a review showed that normal use of e-cigarettes led to fewer harms to respiratory tract than combustible tobacco cigarettes [16]"

This brief comment does not convey the fact that [16] explicitly criticized and highlighted the limitations of the authors' references on respiratory harms. References [12,13,14] are explicitly criticized  in [16]). I explained the criticism of [16] in one full paragraph of the previous report. I am not asking the authors to include all these points, but at the very least they must mention that [16] discussed the limitations of these studies (including [17]). It is adding only one line, but it is an important line that does justice to [16]

(2) Regarding the latest Cochrane review (reference [25]), the authors write:

"Because high-certainty evidence from a recent Cochrane review indicated that e-cigarette use with nicotine aided smokers to quit combustible cigarette smoking more than nicotine replacement therapy [25]",

Again, by stating as result of [25] that e-cigarettes aided more smokers to quit smoking the authors omit the most important result of [25] (concluded by the review):  "There is high certainty that nicotine e-cigarettes increases quit rates compared to nicotine replacement therapy". It is not the same to just aid smokers to quit vs to show high certainty that this happens. The authors must include the sentence "There is high certainty that nicotine e-cigarettes increases quit rates compared to nicotine replacement therapy".

(3) In their cover letter the authors write "I appreciate the comment and agree with you that 18.4% of cigarette smoking is more concerning." Yet they avoid expressing it on the grounds that "the scope of this article is about e-cigarette". This is not a good argument, cigarette smoking and e-cigarette occur jointly and affect each other. However, I truly commend the authors from recognizing that cigarette smoking is the most concerning public health threat and (from the authors' statistics) it is more prevalent in Palestine than e-cigarette usage. The recognition that cigarette smoking is the ultimate harm is shared by all sides in the controversy around e-cigarettes.  I would request the authors just to openly express this concern in their manuscript anywhere they see fit to do it.

There are other points of disagreement with the authors, but I will not add more comments on them. I recognize that the authors have made a valuable attempt to enhance the manuscript.  Once the authors comply with the minot points I mentioned, the article can be published.  

Author Response

 (1) In my previous report I requested the authors to cite reference [16] as a critique of studies on respiratory effects of e-cigarette usage (the authors' references [12,13,14,17]).

Polosa R, O'Leary R, Tashkin D, Emma R, Caruso M. The effect of e-cigarette aerosol emissions on respiratory health: a narrative review. Expert Rev Respir Med. 2019 Sep;13(9):899-915. doi: 10.1080/17476348.2019.1649146.

The authors added the following text commenting on this reference

"While a review showed that normal use of e-cigarettes led to fewer harms to respiratory tract than combustible tobacco cigarettes [16]"

This brief comment does not convey the fact that [16] explicitly criticized and highlighted the limitations of the authors' references on respiratory harms. References [12,13,14] are explicitly criticized  in [16]). I explained the criticism of [16] in one full paragraph of the previous report. I am not asking the authors to include all these points, but at the very least they must mention that [16] discussed the limitations of these studies (including [17]). It is adding only one line, but it is an important line that does justice to [16]
Response: Thank you. Done L 67-74,  but reference 17 was not included in the review [16] since it is more recent [2021].

(2) Regarding the latest Cochrane review (reference [25]), the authors write:

"Because high-certainty evidence from a recent Cochrane review indicated that e-cigarette use with nicotine aided smokers to quit combustible cigarette smoking more than nicotine replacement therapy [25]",

Again, by stating as result of [25] that e-cigarettes aided more smokers to quit smoking the authors omit the most important result of [25] (concluded by the review):  "There is high certainty that nicotine e-cigarettes increases quit rates compared to nicotine replacement therapy". It is not the same to just aid smokers to quit vs to show high certainty that this happens. The authors must include the sentence "There is high certainty that nicotine e-cigarettes increases quit rates compared to nicotine replacement therapy".
Response: Thank you. Done L94

(3) In their cover letter the authors write "I appreciate the comment and agree with you that 18.4% of cigarette smoking is more concerning." Yet they avoid expressing it on the grounds that "the scope of this article is about e-cigarette". This is not a good argument, cigarette smoking and e-cigarette occur jointly and affect each other. However, I truly commend the authors from recognizing that cigarette smoking is the most concerning public health threat and (from the authors' statistics) it is more prevalent in Palestine than e-cigarette usage. The recognition that cigarette smoking is the ultimate harm is shared by all sides in the controversy around e-cigarettes.  I would request the authors just to openly express this concern in their manuscript anywhere they see fit to do it.

Response: Thank you. Added L440-444

  1. English language

Response: Improved.

Reviewer 3 Report

Comments and Suggestions for Authors

The authors made considerable efforts to improve the quality of the manuscript (they are thanked for this). However, Istill have some cncens that should be adresed

1. The first concern is related to the discussion that has not been summarized and contains multiple unnecessary details. You should discuss just the most important results: (exp: the comparison of the rate of response has no sens. Thus, you should delete the paragraph of lines 283-285. You should also delete the paragraph of line 310-315....).

2. The second concern is related to thequality of the English language which reqire an extensive editing (exp: the term e-cigarette is practically repeated in each ln of the introduction...). Revise

Other "minor" comments:

- Delete all the indications to the tables in the methods section.

- Delete the sentence in lines 187-188 "Ex-smokers...regularly".

- Associate table 4 and 5.

- Indicate the significant results in the tables

Comments on the Quality of English Language

The manuscript requires extensive language editing

Author Response

The first concern is related to the discussion that has not been summarized and contains multiple unnecessary details. You should discuss just the most important results: (exp: the comparison of the rate of response has no sens. Thus, you should delete the paragraph of lines 283-285. You should also delete the paragraph of line 310-315....).

Response: Thank you. I removed them and other sentences.

  1. The second concern is related to the quality of the English language which reqire an extensive editing (exp: the term e-cigarette is practically repeated in each ln of the introduction...). Revise

Response: Revised.

Other "minor" comments:

- Delete all the indications to the tables in the methods section.

Response: Done

- Delete the sentence in lines 187-188 "Ex-smokers...regularly".

Response: Done

- Associate table 4 and 5.

Response: Done

- Indicate the significant results in the tables

Response: Thank you. Done

The manuscript requires extensive language editing

Response: Thank you. Revised.